# The force of the myosin motor sets cooperativity in thin filament activation of skeletal muscles

Marco Caremani [1,2], Matteo Marcello[1,2], Ilaria Morotti[1,2], Irene Pertici [1,2], Caterina Squarci[1,2], Massimo Reconditi [1,3], Pasquale Bianco [1,2], Gabriella Piazzesi [1,2], Vincenzo Lombardi [1✉] & Marco Linari [1,2]

Contraction of striated muscle is regulated by a dual mechanism involving both thin, actin-containing filament and thick, myosin-containing filament. Thin filament is activated by $Ca^{2+}$ binding to troponin, leading to tropomyosin displacement that exposes actin sites for inter-action with myosin motors, extending from the neighbouring stress-activated thick filaments. Motor attachment to actin contributes to spreading activation along the thin filament, through a cooperative mechanism, still unclear, that determines the slope of the sigmoidal relation between isometric force and pCa ($-\log[Ca^{2+}]$), estimated by Hill coefficient $n_H$. We use sarcomere-level mechanics in demembranated fibres of rabbit skeletal muscle activated by $Ca^{2+}$ at different temperatures (12–35 °C) to show that $n_H$ depends on the motor force at constant number of attached motors. The definition of the role of motor force provides fundamental constraints for modelling the dynamics of thin filament activation and defining the action of small molecules as possible therapeutic tools.

[1] PhysioLab, University of Florence, Florence, Italy. [2] Department of Biology, University of Florence, Florence, Italy. [3] Department of Experimental and Clinical Medicine, University of Florence, Florence, Italy. ✉email: vincenzo.lombardi@unifi.it

Contraction in striated muscle is driven by myosin motors, extending from the thick filaments and pulling on the neighbouring thin, actin-containing, filaments, through cyclical ATP driven interactions. In the resting muscle, contraction is prevented by tropomyosin (Tm), a regulatory protein in the thin filament that sterically blocks myosin binding sites on actin (Fig. 1).

Nerve action potentials, transmitted via the neuromuscular junction to the muscle cell (fibre), trigger $Ca^{2+}$ release from sarcoplasmic reticulum and rapid activation of the thin filament by $Ca^{2+}$ binding to troponin C (TnC, the $Ca^{2+}$ sensor of the Tn complex, the other regulatory protein in the thin filament) followed by azimuthal displacement of Tm that exposes the actin sites for the interaction with the myosin motors[1]. On the other hand, in the resting muscle most of myosin motors lie on the surface of the thick filament kept folded back towards the centre of the sarcomere (OFF state, Fig. 1) by head-head and head-tail interactions, unable to bind to actin and hydrolyse ATP[2–5]. In loaded contractions the stress generated by a few constitutively ON motors on the thick filament moves the main population of OFF motors away from the thick filament (ON state) making them available for actin binding[6–9]. This process is likely modulated by accessory proteins on the thick filament like the Myosin Binding Protein-C (MyBP-C) or cytoskeleton proteins like titin and their degree of phosphorylation (Fig. 1)[10], but in relaxed demembranated fibres the ordered OFF conformation of the myosin motors is known to be perturbed also by lowering temperature below the physiological value[11–14].

Thin filament activation by $Ca^{2+}$ binding to TnC progresses through the activation of Regulatory Units (RU) composed, for each strand of the double stranded helix, of 7 actin monomers (5.5 nm each in diameter), 1 Tn complex and 1 Tm (Fig. 1). However, $Ca^{2+}$-activation can spread beyond a RU for head-to-tail interactions between consecutive Tm's[15], which suggests a way for the cooperative thin filament activation. Biochemical and structural evidence indicates that full actin site availability implies strong myosin attachment to actin[16–20] supporting the two-step steric-blocking model of thin filament activation:[21] the first step,

triggered by $Ca^{2+}$ binding to TnC, is responsible for a partial displacement of Tm that within a RU leads from the blocked state, in which myosin can only weakly interact with actin, to the closed state that allows weak-to-strong binding transition of myosin; the second step, triggered by strong binding of myosin, promotes further Tm displacement leading to an open state that may in turn provide the cooperative mechanism that spreads thin filament activation[22,23]. Cooperativity determines the slope of the sigmoidal relation between force and pCa in the demembranated fibre, estimated by Hill coefficient $n_H$ (see Methods).

According to a classical view of the role of motors in cooperative activation, for a given number of $Ca^{2+}$-Tn RU's, related to $[Ca^{2+}]$ through the affinity of TnC for $Ca^{2+}$, the value of $n_H$ underlies the number of strongly attached motors (from here on simply called attached motors) and the other way around[22–25]. Accordingly, modulation of the $Ca^{2+}$ affinity by TnC replacement with mutant TnC shifts the force-pCa relation without significant effects on the slope of the relation[26,27]. In this respect, however, a contradiction emerges with the so far unexplained finding that $n_H$ of the force-pCa relations of cardiac and slow skeletal myocytes in the presence of omecamtiv mecarbil (OM) is quite low notwithstanding the high fraction of attached motors[28–30]. OM is an activator of the β/slow isoform of myosin, which increases $Ca^{2+}$ sensitivity by increasing the affinity of myosin for actin but also inhibits the execution of the working stroke and thus force generation and shortening, while the motor remains strongly bound to actin.

Here we use fast sarcomere-level mechanics in $Ca^{2+}$-activated demembranated fibres of rabbit soleus muscle to determine the average force of the motor ($F_0$) under different interventions and define if the stress exerted by the motor on the thin filament has a role in cooperative thin filament activation beyond the steric effect of motor attachment. To this end we study how changing temperature, an intervention that in fast muscle fibres is known to change the force of the myosin motor without changing the number of attached motors[31–34], affects the characteristics of the force-pCa relation. We find that in $Ca^{2+}$-activated fibres of rabbit soleus the rise in temperature from 12 to 35 °C, which does not

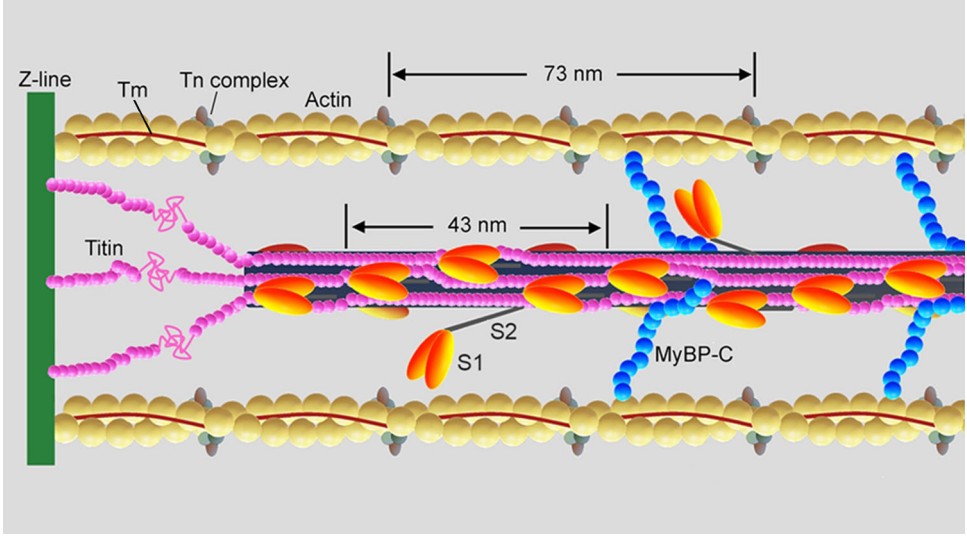

**Fig. 1 Schematic representation of the half-sarcomere protein assembly.** Shown are actin (yellow), tropomyosin (Tm, brown) and troponin complex (Tn, light and dark grey and violet) on the thin filament. On the thick filament (black) most of the S1 globular head domains of myosin (orange) lie tilted back (OFF state) and a few heads (two in the scheme) move away with tilting of their S2 tail domain (ON state); the MyBP-C (blue) lies on the thick filament with the C-terminus and extends to thin filament with the N-terminus. Titin (pink) in the I-band connects the Z line at the end of the sarcomere (green) to the tip of the thick filament and in the A-band runs on the surface of the thick filament up to the M-line at the centre of the sarcomere. Adapted from Fig. 1 in ref. [78] (permission from Springer Nature).

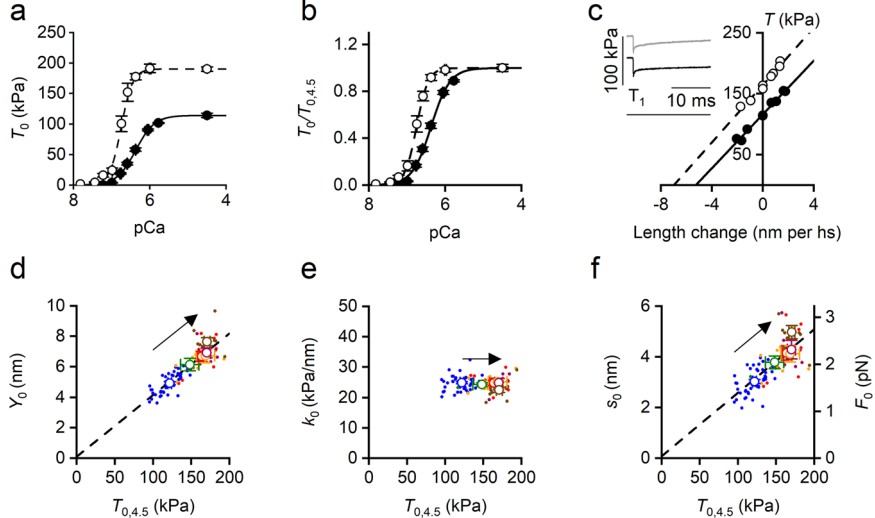

**Fig. 2 Effect of temperature on the relevant mechanical parameters of $Ca^{2+}$-activated skinned soleus fibre. a** Force-pCa relations interpolated with the Hill equation (see SI Methods). Filled circles and continuous line, 12 °C; open circles and dashed line, 35 °C. $T_0$ in absolute units (kPa). **b** As in a with $T_0$ relative to $T_0$ at pCa 4.5 ($T_{0,4.5}$). **c** $T_1$ relations at 12 °C (filled symbols) and 35 °C (open symbols) at pCa 4.5. Lines are first order regression equations fitted to the data at 12 °C (continuous) and 35 °C (dashed). In the inset sample records of the force response to the small step release are superimposed to show the force attained at the end of the step ($T_1$). From top to bottom force at 35 °C, at 12 °C and force baseline. **d** Dependence on temperature -modulated $T_{0,4.5}$ of $Y_0$ (half-sarcomere strain) and fit of pooled data with a first order regression equation (dashed line, slope 0.040 ± 0.003 nm/kPa, ordinate intercept 0.10 ± 0.38 nm). **e** Dependence on $T_{0,4.5}$ of $k_0$ (half-sarcomere stiffness). **f** Left ordinate: dependence on $T_{0,4.5}$ of $s_0$ (motor strain) and linear fit of pooled data (dashed line, slope 0.025 ± 0.003 nm/kPa and ordinate intercept 0.08 ± 0.37 nm). Right ordinate: dependence on $T_{0,4.5}$ of $F_0$ (motor force). The corresponding parameters of the linear fit are 0.014 ± 0.001 pN/kPa (slope) and 0.04 ± 0.20 pN (ordinate intercept). In **d** and **f** the slopes are significantly different from zero ($P$ always <0.005) and the ordinate intercepts are not significantly different from zero ($P$ always >0.75). The arrows in panels **d**–**f** indicate the direction of the rise in temperature from 12 to 35 °C; the colours refer to different temperatures: 12 °C, blue; 17 °C, green; 20 °C, red; 25 °C, orange; 30 °C, purple; 35 °C, brown. In **a** and **b**, data are mean values (± SEM) from 5 fibres. In **c** data from one fibre: fibre length, 3.23 mm; length of the segment under the striation follower, 0.72 mm; average sarcomere length, 2.30 μm; CSA, 4210 μm². In **d**–**f**, filled circles are pooled data and open circles the mean values (± SEM) from 31 fibres.

change the number of attached motors and increases $F_0$ by 50%, increases $n_H$ in proportion. The same increase in temperature in the presence of 1 μM OM, which per se reduces $F_0$ to ½ by preventing the execution of the force-generating step in the OM-bound motors (from here on OM-motors, 50% of the total attached motors[30]), doubles $F_0$, without significant increase in $n_H$. These results suggest that the motor force plays a role in the cooperativity of thin filament activation. Consequently, the classical steric-blocking model must be integrated with the concept that the displacement of Tm and the availability of neighbouring actin sites are under the dynamic control of the motor force. This mechanism is specifically blocked by attachment of motors with force inhibited by OM. Extending the $n_H$–$F_0$ relation in control to higher $F_0$ values with data from fast skeletal muscle reveals that the cooperative mechanism that relates $n_H$ to $F_0$ is unique for the two muscle types, as expected if it specifically depends on the myosin motor force, and saturates at high values of $F_0$ according to a two-step process with Michaelis-Menten kinetics.

## Results

### The effect of temperature on the force-pCa relation and on the force of the myosin motor.
Increase in temperature from 12 to 35 °C increases the isometric force ($T_0$) of the soleus fibre at any $[Ca^{2+}]$, as shown by the force-pCa relations in Fig. 2a (symbols and lines, data and interpolated Hill equation respectively; filled circles and continuous line, 12 °C; open circles and dashed line: 35 °C). The relation at 35 °C is steeper, as it better emerges when $T_0$ is plotted as a fraction of $T_{0,4.5}$ (the maximum value measured at pCa 4.5) at each temperature (Fig. 2b). The underlying cooperativity index (estimated by the parameter $n_H$ of Hill equation, see Methods) increases with temperature and so does

the $Ca^{2+}$-sensitivity of thin filament, as shown by the leftward shift of the pCa at which $T_0$ attains $0.5T_{0,4.5}$ (pCa₅₀, Fig. 2b). In the temperature range 12–35 °C, $T_{0,4.5}$ increases from 127 ± 5 to 183 ± 3 kPa, $n_H$ increases from 1.97 ± 0.07 to 3.09 ± 0.20 and pCa₅₀ increases from 6.38 ± 0.05 to 6.67 ± 0.09 (that is $[Ca^{2+}]_{50}$ reduces from 0.42 μM to 0.21 μM) (Table 1).

According to the simplified mechanical model of the half-sarcomere (see Methods and Supplementary Fig. 1), the half-sarcomere (hs) stiffness of the $Ca^{2+}$-activated fibres, $k_0$, can be used to calculate the strain and force of the motor and the compliance of the myofilaments[33,35]. To this end, step perturbations in length imposed on the fibre at $T_0$ (Supplementary Fig. 2a–c, temperature 12 °C) were used to build the $T_1$ relations at different pCa ranging from 6.8 (Supplementary Fig. 2d, triangles) to 4.5 (circles). The slope and the abscissa intercept of the $T_1$ relations estimate $k_0$ and the hs strain ($Y_0$) respectively. $k_0$ and $Y_0$ increase with the $Ca^{2+}$-dependent increase of $T_0$, though less than in proportion with $T_0$ (Supplementary Fig. 3a, b, respectively), as a consequence of the underlying contributions of two mechanical elements in series (Supplementary Fig. 1): (i) the array of myosin motors, the stiffness of which ($e_0$) increases in proportion to the $Ca^{2+}$-dependent increase of $T_0$ (Supplementary Fig. 3c), underpinning a proportional increase in the fraction of attached motors, and (ii) the myofilaments, the equivalent compliance of which ($C_f$) is estimated as the slope of the $Y_0$-$T_0$ relation (Supplementary Fig. 3b and Eq. 3 in Methods)[33,35].

In the soleus fibre, stiffness measurements can be reliably done at temperatures as high as 35 °C, because the quick force recovery following a step is much slower than in the fast, psoas fibre and the truncation of $T_1$ response by quick recovery is anyway minimum (Fig. 2c). The relation between hs stiffness

**Table 1 Effect of temperature on the relevant mechanical parameters listed on the first row.**

| | Temperature (°C) | $T_{0,4.5}$ (kPa) | $s_0$ (nm) | $F_0$ (pN) | $n_H$ | $pCa_{50}$ |
|---|---|---|---|---|---|---|
| soleus fibre control | | | | | | |
| | 12.3 ± 0.1 (14) | 127 ± 5 | 3.16 ± 0.12 | 1.71 ± 0.06 | 1.97 ± 0.07 | 6.38 ± 0.05 |
| | 17.3 ± 0.1 (4) | 148 ± 12 | 3.78 ± 0.25 | 2.04 ± 0.14 | 2.42 ± 0.09 | 6.54 ± 0.06 |
| | 25.2 ± 0.1 (4) | 165 ± 12 | 4.53 ± 0.20 | 2.45 ± 0.11 | 3.15 ± 0.13 | 6.55 ± 0.03 |
| | 35.5 ± 0.1 (5) | 183 ± 3 | 5.34 ± 0.32 | 2.59 ± 0.19 | 3.09 ± 0.20 | 6.67 ± 0.09 |
| 1 μM OM | | | | | | |
| | 12.2 ± 0.1 (5) | 69 ± 3 | 1.53 ± 0.05 | 0.83 ± 0.06 | 0.79 ± 0.09 | 7.17 ± 0.09 |
| | 25.2 ± 0.1 (5) | 117 ± 14 | 2.75 ± 0.34 | 1.48 ± 0.18 | 0.96 ± 0.13 | 7.21 ± 0.10 |
| | 35.6 ± 0.1 (5) | 134 ± 14 | 3.11 ± 0.30 | 1.68 ± 0.16 | 1.17 ± 0.08 | 7.16 ± 0.08 |
| psoas fibre | | | | | | |
| | 6.0 ± 0.1 (4) | 135 ± 6 | 1.71 ± 0.11 | 2.90 ± 0.18 | 3.51 ± 0.22 | 5.89 ± 0.05 |
| | 12.1 ± 0.1 (12) | 219 ± 11# | 2.86 ± 0.16# | 4.87 ± 0.27 | 3.92 ± 0.21 | 6.00 ± 0.13 |
| | 25.0 ± 0.1 (6) | 317 ± 14# | 4.13 ± 0. 19# | 7.02 ± 0.32 | 4.81 ± 0.22 | 6.46 ± 0.06 |

Data for soleus fibres in control and in the presence of 1 μM OM. For comparison also data for psoas fibres are reported (from ref. [33] and new data). Data are mean ± SEM. In brackets the number of fibres used for each temperature. # mean calculated by pooling data from new experiments with data from ref. [33]. In each of the three cases the statistical significance of the temperature effect is tested with one-way ANOVA as detailed in Supplementary Table 2. The soleus values in control of $F_0$ and $n_H$ at 35.6 °C are not significantly different from the corresponding psoas values at 6.0 °C ($P = 0.24$ and 0.16 respectively).

and $Ca^{2+}$-dependent force and the ensuing analysis of the half-sarcomere compliance are repeated at 35 °C and are super-imposed on those at 12 °C in Supplementary Fig. 4a–c (filled circles 12 °C, open circles 35 °C). The analysis makes evident that the motor strain $s_0$ (the ordinate intercept of the $Y_0$–$T_0$ relation, Supplementary Fig. 4b) is the parameter that accounts for both the 40% increase in force between 12 and 35 °C and the apparent reduction, at a given force, of the stiffness of the half-sarcomere (Supplementary Fig. 4a) and the motor array (Supplementary Fig. 4c).

The mechanism underlying the potentiating effect of temperature on $T_{0,4.5}$ is analysed at six temperatures in the range 12–35 °C in Fig. 2d–f. It can be seen that the increase in $T_{0,4.5}$ is accounted for by a proportional increase in hs strain ($Y_0$, Fig. 2d) without significant change in hs stiffness ($k_0$, 24.4 ± 0.4 kPa/nm estimated in the whole temperature range, Fig. 2e). Consequently, also $e_0$, obtained according to Eq. 2 by subtracting from the hs compliance ($1/k_0$) the contribution of filament compliance ($C_f$, 13.66 ± 3.02 nm/MPa, estimated in the whole temperature range as in Supplementary Fig. 3b) is constant independent of temperature (39.9 ± 0.9 kPa/nm), indicating that the fraction of the attached motors does not change with temperature. $s_0$, instead, increases in proportion to $T_{0,4.5}$ (Fig. 2f, left ordinate) as shown by the first order equation fit to data (line) that exhibits an ordinate intercept not significantly different from zero.

It has been previously shown that under the same experimental conditions used here (temperature 12 °C, sarcomere length 2.3–2.5 μm, ionic strength 190 mM) the fraction of attached motors at $T_{0,4.5}$ in soleus fibres is 0.47[35]. With this value and the known density of myosin motors per half thick filament ($1.58*10^{17}$ m$^{-2}$[35],) the stiffness of the attached myosin motor ($\varepsilon_0$) can be calculated from $e_0$ and is $[39.87/(0.47*1.58*10^{17}$ m$^{-2}$) =] 0.54 ± 0.01 pN/nm. The average force per myosin motor $F_0$ can be calculated at each temperature by the product $s_0*\varepsilon_0$. $F_0$ increases with $s_0$ from 1.7 pN at 12 °C to 2.6 pN at 35 °C (Fig. 2f, right ordinate and Table 1).

**The effect of temperature in the presence of 1 μM OM.** In the presence of 1 μM OM the rise in temperature from 12 to 35 °C increases the isometric force of the soleus fibre at any $[Ca^{2+}]$ and the slope of the force-pCa relation (Fig. 3a and b: 12 °C, filled circles and interpolated black continuous line; 35 °C, open circles and interpolated black dashed line; in b, data are plotted relative to $T_{0,4.5}$ at each temperature). At either temperature, in OM with respect to control (grey lines from black lines in Fig. 2a and b

respectively), $T_{0,4.5}$ is depressed, $n_H$ is smaller and pCa$_{50}$ is larger (see also Table 1). Moreover, $n_H$ increases with temperature (12–35 °C, $P < 0.05$) in OM as in control, while pCa$_{50}$ does not change significantly ($P > 0.9$, Table 1).

Stiffness measurements in the presence of 1 μM OM (Fig. 3c) show that, as in control, the slope of the $T_1$ relation at $T_{0,4.5}$ is the same at 12 °C (filled circles and continuous line) as at 35 °C (open circles and dashed line). The half-sarcomeres strain, $Y_0$, determined at $T_{0,4.5}$ at five temperatures in the range 12–35 °C, increases in proportion with $T_{0,4.5}$ (Fig. 3d: circles, data points and black line, first order equation fit), like in control (grey line from Fig. 2d), so that the half-sarcomere compliance (the slope of the relation, 39.7 ± 2.6 nm/MPa) is not significantly different from that in control (40.5 ± 2.6 nm/MPa, Fig. 2d, $P > 0.8$). Consequently, the hs stiffness, $k_0$, is constant in the whole temperature range (Fig. 3e, average 24.6 ± 0.6 kPa/nm) and not significantly different from that in control (Fig. 2e, $P > 0.7$).

$s_0$ and $C_f$ at any given temperature have been estimated in the presence of 1 μM OM by applying the same analysis as in control (see Supplementary Fig. 3d for temperature 12 °C and Supplementary Fig. 4d for the effect of rising temperature to 35 °C) to hs stiffness measurements in fibres activated at different $[Ca^{2+}]$: $s_0$ increases from 1.66 ± 0.14 nm to 3.03 ± 0.28 nm with temperature while $C_f$ is constant.

The effect of temperature on $T_{0,4.5}$ in the range 12–35 °C is explained by proportional changes in $s_0$ (Fig. 3f, left ordinate), while the stiffness of the motor array $e_0$, calculated from $k_0$ (Fig. 3e) by subtracting the contribution of filament compliance, is independent of temperature (39.1 ± 0.6 kPa/nm) and practically identical to the value in control (39.9 ± 0.9 kPa/nm), indicating that in either condition the fraction of attached motors underpinning $T_{0,4.5}$ is the same as reported in ref. [30]. The fraction of attached motors that are bound to OM (50% in 1 μM OM at 12 °C according to ref. [30]) does not generate force[29,30]. Thus, in 1 μM OM at 12 °C, while the total fraction of attached motors, estimated by $e_0$, is the same as in control, $T_{0,4.5}$ (69 ± 3 kPa) is ½ of that in control (127 ± 5 kPa, Table 1) under the condition that ε is the same[30]. Temperature dependent changes of $T_{0,4.5}$ in either case are totally accounted for by proportional increases in average $s_0$ (Fig. 3f left ordinate, compare grey (control) and black (OM) dashed lines) and thus in $F_0$ (= $\varepsilon*s_0$, right ordinate).

**The relation between $n_H$ and motor force.** Both changing temperature and addition of OM are effective interventions for

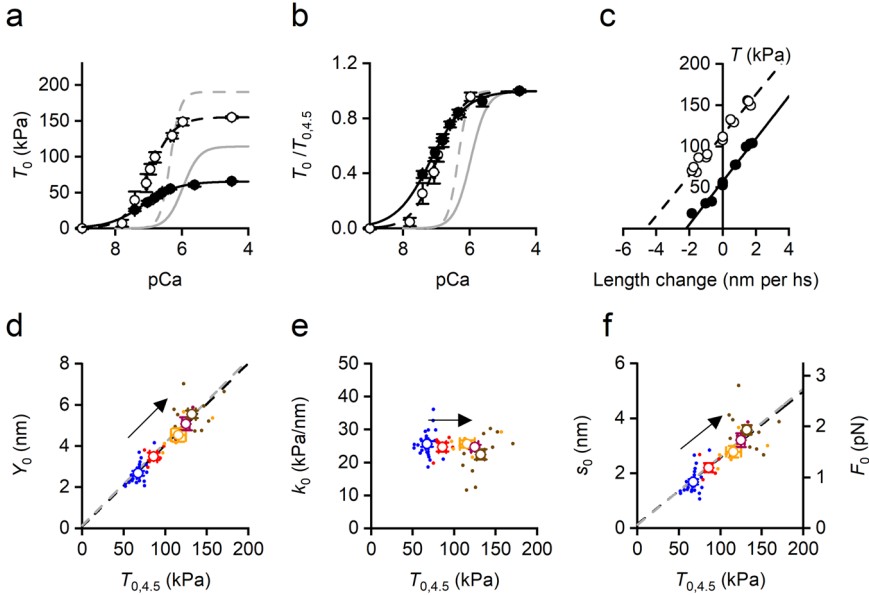

**Fig. 3 Effect of temperature on the relevant mechanical parameters of Ca²⁺-activated skinned soleus fibre in the presence of 1 μM OM.** OM relations (black symbols and lines) are compared with relations in control from Fig. 2 (grey lines). **a** Force-pCa relations interpolated with the Hill equation at 12 °C (filled circles and continuous line) and 35 °C (open circles and dashed line). $T_0$ in absolute units (kPa). **b** As in **a** with $T_0$ relative to $T_0$ at pCa 4.5 ($T_{0,4.5}$). **c** $T_1$ relations at 12 °C (filled symbols) and 35 °C (open symbols). Lines are first order regression equations fitted to the data at 12 °C (continuous) and 35 °C (dashed). **d** Dependence on temperature modulated $T_{0,4.5}$ of $Y_0$ (half-sarcomere strain,) and fit of pooled data with a first order regression equation (dashed line, slope 0.040 ± 0.003 nm/kPa, ordinate intercept 0.09 ± 0.27 nm). **e** Dependence on $T_{0,4.5}$ of $k_0$ (half-sarcomere stiffness). **f** Left ordinate: dependence on $T_{0,4.5}$ of $s_0$ (motors strain) and linear fit of pooled data (dashed line, slope 0.024 ± 0.003 nm/kPa and ordinate intercept 0.16 ± 0.26 nm). Right ordinate: dependence on $T_{0,4.5}$ of $F_0$ (motor force). The corresponding parameters of the linear fit are 0.013 ± 0.001 pN/kPa, slope and 0.08 ± 0.14 pN, ordinate intercept. In **d** and **f** the slopes are significantly different from zero (P always <0.005) and the ordinate intercepts are not significantly different from zero (P always >0.5). The arrows indicate the direction of the rise in temperature from 12 to 35 °C; the colours refer to different temperatures: 12 °C, blue; 20 °C, red; 25 °C, orange; 30 °C, purple; 35 °C, brown. In **a** and **b**, data are mean values (±SEM) from 7 fibres. In **c** data from one fibre: fibre length, 4.75 mm; length of the segment under the striation follower, 0.63 mm; average sarcomere length, 2.31 μm; CSA, 2800 μm². In **e** and **f** filled circles are pooled data and open circles the mean values (±SEM) from 17 fibres.

modulating the isometric force of Ca²⁺-activated fibres of rabbit soleus through mechanisms that change the average force of the myosin motor without changing the number of attached motors. Under this condition it is possible to test the specific role of the force of the motor in the cooperative activation of the thin filament without the confounding effect that changes in number of attached motors could introduce through the steric-blocking mechanism that links motor attachment and Tm displacement.

Temperature-dependent changes of $n_H$ in the control force-pCa relations are linearly related to $F_0$ (Fig. 4a: filled circles pooled data, open circles average at each temperature; blue 12 °C, green 17 °C, orange 25 °C, brown 35 °C). The first order equation fit to pooled data (continuous line) gives a slope of 1.10 ± 0.15 $n_H$ units pN⁻¹. In OM the $n_H$-$F_0$ relation in the same range of temperatures (triangles, same colour code as control) exhibits a much smaller sensitivity on $F_0$ (the linear fit to pooled data, dashed line, gives a slope of 0.25 ± 0.13 $n_H$ units pN⁻¹) and is shifted downward by at least 1 $n_H$ unit.

The pCa₅₀ in the same temperature range appears to be related to $F_0$ in control (Fig. 4b, circles, same symbols and colour code as in a) but not in OM (triangles), in which case the data are shifted upward by at least 0.5 pCa units (see also Table 1). The much larger Ca²⁺ sensitivity characterising the force-pCa relation in OM is in agreement with structural evidence from both fluorescent probes[36] and X-ray diffraction[37] that OM binding to myosin promotes the release of a fraction of motors from the resting OFF state even at low [Ca²⁺]. The increase of pCa₅₀ in OM is accompanied by a marked reduction in cooperativity, shown by the downward shift of $n_H$ data and blunting of the $n_H$ sensitivity to $F_0$ (triangles in Fig. 4a). These features make evident

that the inhibitory effect of OM on $n_H$ is beyond that expected from the reduction of the average value of $F_0$ and must be explained by the specific mechanism by which the average $F_0$ is reduced in OM with respect to control, namely the loss of force generating capability by OM-motors[29,30].

**A mechanistic model that explains the dependence of cooperativity on $F_0$ and the inhibition of cooperativity by OM.** The qualitative model presented here serves as a proof-of-concept for the hypothesised mechanism by which $F_0$ determines the degree of cooperativity of thin filament activation. To keep the model as simple as possible (i) the 3D lattice of thin and thick filaments at full filament overlap is reduced to one thick filament facing a single stranded thin filament and the number of myosin dimers per RU is assumed to be two according to the calculation detailed in Supplementary Note 1, (ii) 12 RU's are considered as a minimum representative number, and (iii) the description of the events is limited to a [Ca²⁺] (10⁻⁷ M) just above the threshold of the force-pCa relation (the minimum [Ca²⁺] for a detectable rise of active force), at which only two RU's are assumed to be activated by Ca²⁺.

The basic assumptions concerning the cooperative mechanism and its inhibition by OM are (i) the longitudinal extent of Tm displacement and thus the number of neighbouring actin monomers that become available ($n_A$) on either side of the monomer strongly bound to a motor depend on motor force $F$, which corresponds to the average motor force $F_0$ determined experimentally in control (Table 1) and, at a given temperature, is the same for all force generating motors, independent of

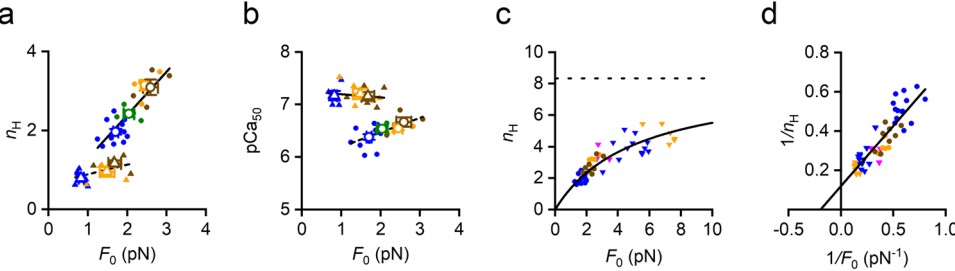

**Fig. 4 Dependence of $n_H$ and pCa$_{50}$ on $F_0$. a** $n_H$–$F_0$ relations for the soleus fibres in control (circles) and in the presence of 1 µM OM (triangles). Filled symbols, pooled data from twenty-three fibres; open symbols, mean values (±SEM) at each temperature. Different colours refer to different temperatures: 12 °C, blue; 17 °C, green; 25 °C, orange; 35 °C, brown. Lines are first order regression equations fitted to pooled data (continuous, control; dashed, OM). The slope and the ordinate intercept of the regressions are: control, 1.10 ± 0.15 pN$^{-1}$ and 0.19 ± 0.32 respectively; 1 µM OM, 0.25 ± 0.13 pN$^{-1}$ and 0.65 ± 0.18 respectively. **b** pCa$_{50}$–$F_0$ relations in the soleus fibre in control (circles) and in the presence of 1 µM OM (triangles) from the same force-pCa relations as in a. Lines are the first order regression equations fitted to pooled data (continuous, control; dashed, OM). The slope and the ordinate intercept of the regressions are: control, 0.28 ± 0.07 pN$^{-1}$ and 5.92 ± 0.14 respectively; 1 µM OM, −0.06 ± 0.11 pN$^{-1}$ and 7.26 ± 0.15 respectively. **c** Superimposed $n_H$–$F_0$ relations in control from the soleus fibres (circles from **a**) and from psoas fibres (reverse triangles, pooled from ref. [33] and new data for a total of twelve fibres). Different colours of reverse triangles refer to different temperatures: 6 °C, magenta; 12 °C, blue; 25 °C, orange. The continuous line is calculated from the linear fit of the relation of reciprocal of pooled $n_H$–$F_0$ data (continuous line in **d**), the horizontal dashed line is drawn starting from the ordinate that is the reciprocal of the ordinate intercept in **d**. **d** Same data as in **c** plotted as reciprocals. The line is the first order regression equation fitted to the pooled data (slope, 0.61 ± 0.05 pN; ordinate intercept, 0.12 ± 0.02; abscissa intercept -0.20 ± 0.04 pN$^{-1}$).

the presence/absence of OM in the solution. $n_A$ depends on $F$ according to the expression $n_A = 4 + 2.6*F$. At 12 °C ($F = 1.7$ pN) $n_A \sim 8$ and at 35 °C ($F = 2.6$ pN) $n_A \sim 11$. OM-motors have $F = 0$ and thus, following the attachment of an OM-motor, $n_A = 4$. (ii) a motor in the ON state attaches to the nearby actin monomer if it is made available by Tm displacement caused by either Ca$^{2+}$-activation of the RU or the cooperative action of a nearby attached motor; (iii) the attachment of an OM-motor in a region where also other attached motors give their contribution to thin filament activation inhibits the action of the force generating motors, limiting Tm displacement and $n_A$ according to the action of the OM-motor.

Under the conditions of the present experiments, we assume that thin filament activation and its cooperativity are not influenced by mechanosensing-based myosin filament activation[6]. In fact, the temperature jump technique exploited to elicit contractions preserving the sarcomere order and the possibility to apply fast mechanical methods to a selected population of sarcomeres requires that the activating solution is first equilibrated into the fibre at 1 °C[33], at which the myosin filament structure of the relaxed skinned fibres has already undergone a substantial transition to the ON/disordered state[14]. The absence of a significant effect of myosin filament activation in the present experiments appears evident from the lack of asymmetry in the force-pCa relation (Fig. 2b and Supplementary Fig. 5), which underpins only one mechanism for cooperativity in actin filament activation[38], as explained in detail in Discussion. Accordingly, the fraction of motors that in control is able to attach to the activated actin monomers at [Ca$^{2+}$] just above the threshold in Fig. 5a, b is assumed constant and $= (6/24) \frac{1}{4}$, only a bit less than the fraction attached in the steady isometric contraction as suggested by the classical duty ratio value (~1/3). The further final progression in myosin filament activation due to OM (Fig. 5c, d) is assumed to explain the leftward shift of the threshold of the force-pCa relation. In this respect the finding that in control solution the threshold does not change with the test temperature (Fig. 2b, but see also ref. [24]) confirms that the degree of myosin filament activation is pre-determined by the low temperature before the temperature-jump activating protocol.

The dependence of cooperativity on $F_0$ in control is explained by the model by comparing the sequence of events at 12 °C

(Fig. 5a) and 35 °C (Fig. 5b) at pCa 7, at which only two RU's of the twelve considered are Ca$^{2+}$-activated (line 1). The step-by-step process at either temperature is described in detail in Supplementary Note 1. The final outputs are that two motors are attached and develop force at 12 °C (line 3 in a), while at 35 °C three motors are attached and develop force (line 4 in b).

The explanation (at 35 °C) of the double action of OM, that is the leftward shift of the threshold of the force-pCa relation with increase in pCa$_{50}$ and the drop in $n_H$, emerges from the sequence of events described in Fig. 5c and d. The increased spatial frequency of all motors in the ON state for the contribution of OM-motors anticipates the attachment/force generation by the first motor to pCa 7.7 (Fig. 5c), at which only one RU is activated (line 1). In this way OM induces a leftward shift of the threshold of the force pCa-relation that accounts for the observed increase in pCa$_{50}$ (Fig. 3b). At pCa 7 (Fig. 5d), a sub-saturating pCa at which the number of activated RU's is 2 (line 1) and the observed force-pCa points in OM approach those in control (Fig. 3b), $n_H$ is reduced by the interposition of zero force OM-motors which specifically limit the propagation of thin filament activation by force-generating motors, as described in detail in Supplementary Note 1. The final output is that in the presence of OM, at pCa 7 and 35 °C, the steady state force is that of two motors (line 3 in d), less than in control at the same pCa and temperature (3 motors, line 4 in b), in spite of the leftward shift of the threshold.

## Discussion

**The relation between $n_H$ and $F_0$.** We find that, in Ca$^{2+}$-activated demembranated fibres of slow skeletal muscle (rabbit soleus), under conditions in which the number of strongly bound myosin motors is kept constant, the cooperativity in thin filament activation expressed by Hill coefficient $n_H$ is linearly related to the force of the myosin motor $F_0$. Both $n_H$ and $F_0$ are relatively small compared to the corresponding values in fast skeletal muscle of the same animal (rabbit psoas) expressing the fast Heavy Mero Myosin (HMM) isoform, which develops a force $F_0$ up to 7 pN at near physiological temperature[33,35]. The superposition of data from the two muscle fibres (Fig. 4c, soleus: circles, pooled data from Fig. 4a; psoas: reverse triangles, pooled data from ref. [33] and new experiments) provides striking further elements in support of the finding that cooperativity depends on the motor force

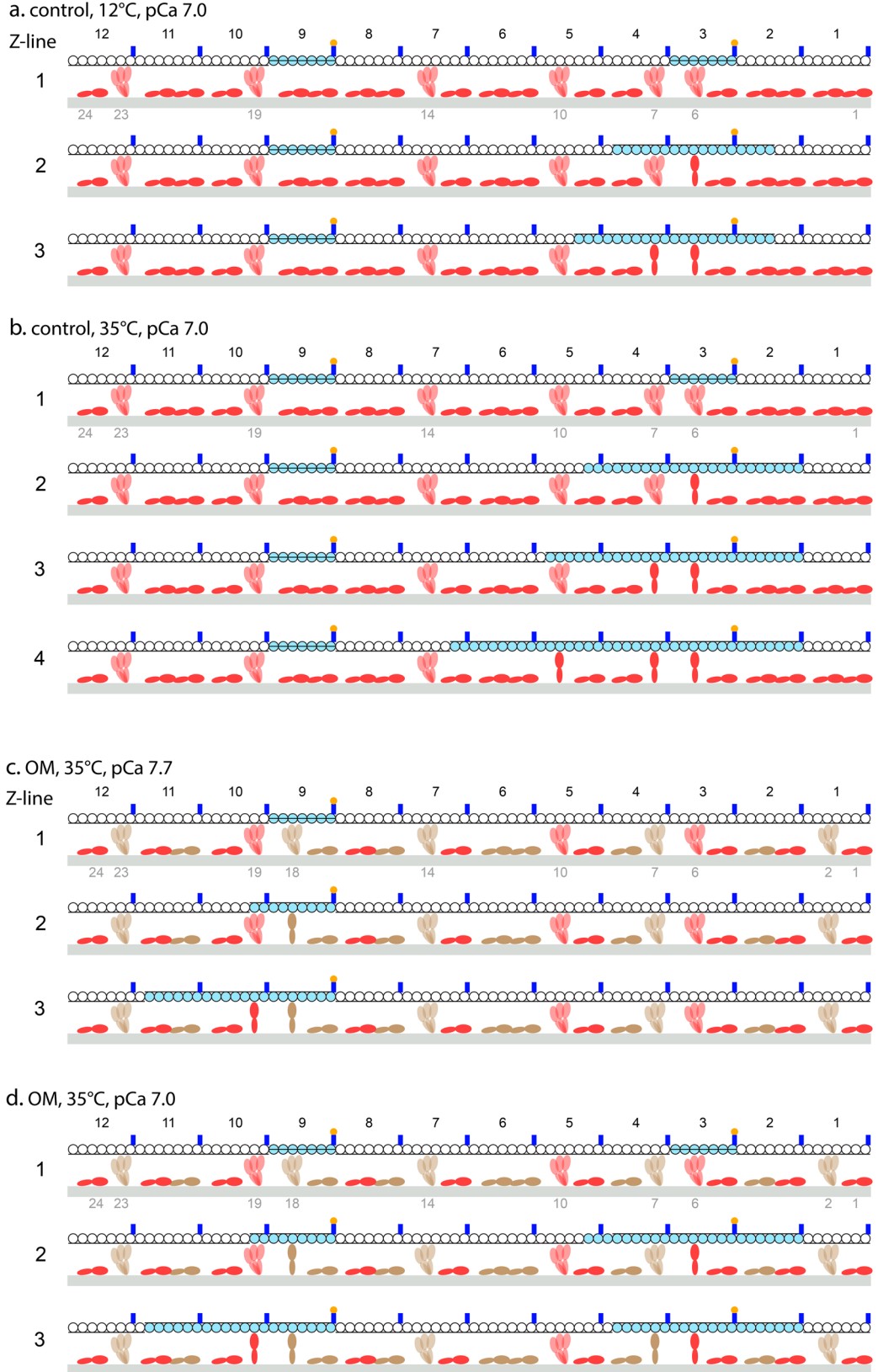

allowing the identification of the underlying molecular mechanism. The first element is that the $n_H$-$F_0$ relation of fast muscle appears as the continuation, in the higher $F_0$ range, of the same relation as that of slow muscle, indicating that the mechanism encompasses the muscle type, relating $n_H$ to the motor force per se. Notably, the mean $n_H$-$F_0$ point of the soleus fibre at 35 °C

is not significantly different from the corresponding point of the psoas fibre at 6 °C (Table 1), further solidifying the idea that the only determinant of $n_H$ is $F_0$, without any further effect of temperature or protein isoform. As a corollary it can be seen that the corresponding $n_H - T_0$ points belong to different relations depending on the myosin isoform (Supplementary Fig. 6, circles

**Fig. 5 Mechanistic model to explain the dependence of the cooperativity ($n_H$) on the force per myosin motor ($F_0$) and the inhibitory effect of OM.** In each line of **a**–**d**: the upper filament is the thin filament with monomers (circles) either blocked (white) or available for myosin attachment (cyan), Tm, black line shifted according to the three states (down blocked, middle closed, up open). Tn complex, blue, $Ca^{2+}$, yellow. The filament is reduced to 12 RU's for simplicity (identified by black numbers from 1 to 12 in the Z-ward direction); the lower filament is the thick filament (grey) carrying two myosin molecules per RU (red without OM, brown OM-bound) identified by green numbers from 1 to 24 in the Z-ward direction. The 18 molecules lying on the surface of the thick filament represent those not available for actin attachment, the 6 that emerge from the backbone are those that attach to an actin site once it is made available by Tm displacement (black line moves up) caused by either $Ca^{2+}$- activation of the RU or the cooperative action of a nearby attached motor. In each panel, line 1 represents the starting conditions at a given pCa and the subsequent lines represent the sequence of events triggered by activation and myosin attachment until a steady state for that pCa is attained. **a** and **b**, in control solution at pCa 7 and 12 °C and 35 °C respectively. **c** and **d**, in the presence of 1 μM OM at 35 °C and pCa 7.7 and 7 respectively.

soleus, reverse triangles psoas). The psoas $n_H - T_0$ point at 6 °C is leftward shifted with respect to the soleus $n_H - T_0$ point at 35 °C, with which it shares the same $F_0$, because of the isoform-dependent effect on $T_0$[35]. In fact, the isometric duty ratio is higher in soleus than in psoas and when the comparison is made at the respective temperature at which $F_0$ is the same, soleus $T_0$ is higher than psoas $T_0$ (Table 1).

The second element is that $F_0$ of fast muscle attains values at which the cooperative mechanism shows a saturation kinetics that suggests a Michaelis-Menten reaction. Indeed, when $n_H$-$F_0$ data are plotted as reciprocals (Fig. 4d), the $1/n_H$–$1/F_0$ relation can be fitted by a first order equation (continuous line) that, according to the Michaelis-Menten kinetics, gives an ordinate intercept that estimates $1/n_{H,max}$ (the reciprocal of the maximum value of $n_H$ attained at saturating values of $F_0$) and an abscissa intercept that estimates $-1/K_M$ (the negative reciprocal of $K_M$, the value of $F_0$ at which $n_H$ is half-maximum). $n_{H,max}$ and $K_M$ are 8.34 ± 1.41 and 5.12 ± 0.95 pN respectively. Notably $K_M$ is larger than the maximum $F_0$ achieved in soleus fibres, which explains the apparent linearity of the $n_H$-$F_0$ relation in this muscle with the evidence that its occupancy of the whole $n_H$-$F_0$ relation is limited to the lower branch of the hyperbola.

Increase in temperature also induces increase in pCa$_{50}$ (circles in Fig. 4b; see also refs. [24,39–41]). Obviously, whenever the $n_H$-$F_0$ points belong to force-pCa relations that share the same threshold, as it occurs when the force-pCa is modulated by temperature (Fig. 2b and ref. [24]), an increase in $n_H$ necessarily underpins an increase in pCa$_{50}$. The correlation between $n_H$ and pCa$_{50}$ is lost when extended to different fibre types, as demonstrated by the finding that the same force-pCa relations of the psoas fibres holding the highest $n_H$ values in Fig. 4c (reverse triangles) and Table 1 hold the smallest pCa$_{50}$ values in Table 1. This can be explained considering that the $Ca^{2+}$-sensitivity of the thin filament, and thus the threshold pCa and pCa$_{50}$ depend primarily on the TnC isoform, while cooperativity does not: the slow skeletal/cardiac TnC has higher $Ca^{2+}$ affinity and larger pCa$_{50}$ than the fast skeletal TnC[42–44], but lower cooperativity as expected by its dependence on $F_0$.

**Considerations in relation to previous work.** The association of the rightward shift of the force-pCa relation with the increase in $n_H$ in fast fibres with respect to slow fibres could indicate a relation between reduction in $Ca^{2+}$ sensitivity and increase in cooperativity. In this respect, however, it must be noted that there is a systematic association between the isoforms of TnC and HMM: fast muscle fibres with fast MHC express the fast TnC, while slow muscle fibres with the slow MHC isoform express the slow TnC isoform[45]. The larger $n_H$ of the fast muscle fibres can be explained by the presence of the fast MHC isoform, which develops a higher $F_0$, not dependent on the TnC isoform. The co-presence of the fast TnC isoform explains per se the rightward shift of the force-pCa relation. This conclusion agrees with previous works[26,27] showing that, in fast skeletal muscle fibres, the

replacement of native TnC with TnC mutated for its affinity for $Ca^{2+}$ produced shift of the pCa$_{50}$ to either the left (when the $Ca^{2+}$ affinity was increased) or the right (when the $Ca^{2+}$ affinity was reduced) without significant effect on the cooperativity. The above conclusion is challenged by results of experiments on fast skeletal muscle myofibrils in which the fast isoform of the Tn complex was replaced with the slow/cardiac isoform[46]. The substitution produced a leftward shift of pCa$_{50}$ accompanied by a marked reduction of $n_H$ from 3.3 to 1.4, that is a value even smaller than that found in general in cardiac myocytes or myofibrils (~3[47],). Notably the replacement of the fast TnC isoform with the slow/cardiac isoform did not affect significantly either pCa$_{50}$ or $n_H$, even if it reduced the mechanical performance (maximum isometric force, rate of force development). That the issue for the slow/cardiac muscle is more complicated than for the fast skeletal muscle is demonstrated by the finding that in cardiac myofibrils, in contrast to the finding in fast skeletal myofibrils[26,27], the replacement of native cardiac TnC (cTnC) with cTnC mutated for its affinity for $Ca^{2+}$ produced shift of the pCa$_{50}$ to either the left (when the $Ca^{2+}$ affinity was increased) or the right (when the $Ca^{2+}$ affinity was reduced), in both cases accompanied by the reduction of $n_H$[48]. This peculiar response of cTnC substitution in cardiac myofibrils could be related to the reported evidence that in cardiac muscle motor attachment more strongly affects cooperativity increasing either $Ca^{2+}$ binding to cTnC per se or the action of cTnC in the regulatory complex[49–51]. The finding in this work that cooperativity depends on the force of the motor suggests that a substantial contribution to solve the question would be to apply the half-sarcomere compliance analysis shown in this paper to test whether and how the replacement of the regulatory proteins changes the force of the myosin motor.

Previous works showed that the slope of the force-pCa relation at forces below the midpoint is higher than that at forces above the midpoint[24,52,53]. This asymmetry is expected if at low $[Ca^{2+}]$ more than one mechanism, as for instance the recently discovered mechanosensing-based myosin filament activation[6], contributed to cooperativity. In fact, while the thin filament activation is complete only with $Ca^{2+}$ saturation at the maximum force ($T_{0,4.5}$), mechanosensing-based myosin filament activation is already complete when the force is nearly half-maximum[8]. Thus, where present, the effect of partial myosin filament activation could affect the shape of the first half of the force-pCa relation, making the whole relation asymmetric across the force midpoint and weakening the solidity of our analysis that is based on the explanation of the slope of the relation only with the thin filament cooperative activation. In this respect, a crucial finding in our work is that, independently of temperature, the force-pCa relation for the soleus fibre is symmetric across the force midpoint (Fig. 2b, filled circles 12 °C, open circles 35 °C). A more stringent test for the presence of asymmetry is obtained with the linearization of the force-pCa relation by expressing the force as $\log [T_0/T_{0,4.5}/(1 - T_0/T_{0,4.5})]$, The linear plots do not show change in the slope across the zero ordinate, corresponding to the

force midpoint (Supplementary Fig. 5, filled circles 12 °C, open circles 35 °C), as it is expected from a symmetric force-pCa relation. Considering the previous finding in our lab that also the psoas fibre exhibits a symmetric force-pCa relation[54], we can conclude that in both muscle types there is only one cooperativity mechanism throughout the whole pCa range[38].

The absence in our experiments of a significant effect of myosin filament activation on the lower half of the force-pCa relation is likely due to the use of the temperature jump technique for eliciting the contraction. The method is essential to avoid the sarcomere disorder caused by the time for $Ca^{2+}$ equilibration within the fibre and preserve the possibility to apply fast mechanical methods to a selected population of sarcomeres[33]. As demonstrated in ref. [14], the myosin filament structure of relaxed skinned fibres has already undergone a substantial transition to the ON/disordered state at 10 °C and presumably this transition is even more effective at 1 °C.

In works of other laboratories, in which contractions are elicited by directly shifting the fibre from low- to high- $[Ca^{2+}]$ solution, the time needed for $[Ca^{2+}]$ equilibration within the fibre implies that force development is accompanied by increase in sarcomere disordering. This per se can provide an explanation for the reported asymmetry in the force-pCa relation since sarcomere disordering is larger at larger forces (smaller pCa) and thus more effectively depresses the upper half of the force-pCa relation. Moreover, the consequences of $Ca^{2+}$ diffusion time are likely worse in fast fibres versus slow fibres, due to the faster rate of force development, which makes more effective the delay in $[Ca^{2+}]$ equilibration. This would provide an explanation for the finding that the maximum isometric force ($T_{0,4.5}$) has been reported to be similar in slow and fast fibres[24], while in our experiments it is 70–90% higher in fast than in slow fibres (Table 1, but see also ref. [35]).

Our high-resolution sarcomere-level mechanics allowing in situ measurement of the force per motor provides a key for explaining the apparent contradiction that changes in $T_0$ are found both related and unrelated with $n_H$. We demonstrate that, when changes in $T_0$ of a given muscle type are completely explained by changes in $F_0$ (as it occurs by changing temperature in soleus fibres), the same relation to $n_H$ holds for either $F_0$ or $T_0$ (Supplementary Fig. 6, open and filled circles). On the contrary, other interventions that induce depression in $T_0$ as the raise in orthophosphate (Pi) or vanadate (Vi) concentrations have been found not to affect significantly the cooperativity parameter of the force-pCa relation[24,55]. According to the present work, those results find a straightforward explanation in the demonstration that the reduction of $T_0$ obtained by increasing either Pi[56] or Vi[57] is due to a proportional reduction in the number of attached motors, without effect on $F_0$.

**Revision of the classical steric-blocking model and the effect of OM on cooperativity.** The $n_H$-$F_0$ relation determined in our experiments demonstrates that the classical two step steric-blocking model[21] must be implemented with the introduction of the concept of the dynamic nature of the movement of Tm promoted by motor attachment: the amplitude of the azimuthal movement of Tm and its axial extension, which should represent the intermediate step in terms of Michaelis-Menten kinetics, depend on $F_0$ with a saturation kinetics and so does the number of cooperatively activated actin monomers (the final product of the reaction).

The intermediate reaction hypothesised above assumes the presence of a link between force development and azimuthal movement of Tm, which may find support in mechanical and structural evidence that the temperature-dependent increase in

the isometric force is related to a higher free energy change between different conformations of the attached myosin motors that shifts the equilibrium to higher force generating states[31,32,58,59] and likely implies an azimuthal reorientation of the motor on the actin monomer it is bound to[60,61]. Further structural evidence for an azimuthal component of the working stroke that promotes Tm displacement can be found in cryo-EM images of isometrically contracting Insect Flight Muscle[62]. Alternatively, it could be argued that it is the Tm that modulates the generation of force by myosin in relation to either temperature or its slow and fast isoform expression in slow and fast muscle fibres respectively.

Addition of OM induces a drastic reduction in $n_H$ and in the $n_H$-dependence on average $F_0$, which cannot be accounted for by the reduced value of $F_0$ (Fig. 4a, triangles). The explanation resides in the specific mechanism by which the average $F_0$ is reduced by OM: while reduction of the average $F_0$ by reducing temperature is accounted for by shifting of the whole population of attached motors towards lower force generating states of the working stroke[31], the reduction of the average $F_0$ by OM is attributed to arresting the force generating mechanism in OM-motors[29]. In this way an OM-motor would counteract the force-dependent Tm displacement by a neighbour force-generating motor, causing a direct inhibition of cooperativity.

The mechanisms underlying the dynamic nature of the action of the myosin motor on thin filament cooperative activation and the dual effect of OM (increase in $Ca^{2+}$ sensitivity and inhibition of cooperativity) are qualitatively tested with a simplified representation of the sequence of the events occurring just beyond the threshold for $Ca^{2+}$ activation (Fig. 5), providing fundamental constraints for the implementation of a quantitative model able to give a detailed description of the role of the motor force on the dynamics of thin filament activation and for the definition of the effects of small molecules as possible therapeutic tools.

## Methods

**Animals and ethical approval.** Experiments have been done on demembranated fibres from the soleus muscle of adult male New Zealand white rabbits at the PhysioLab Research Unity of the Biology Department of the University of Florence. Some additional experiments, beyond those already published in ref. [33], were done also on demembranated fibres from the psoas muscle of the rabbit. The experiments were carried out according to the protocols approved by the Ethical Committee of the University of Florence and by the Italian Ministry of Health (authorization n. 956/2015 PR) in compliance with the Italian regulation on animal experimentation, Decreto Legislativo 26/2014 and the EU regulation (directive 2010/63) (authorization n. 55FF7.N.THO). Rabbits (4–5 kg weight, 20–30 weeks old) were sacrificed by injection of an overdose of sodium pentobarbitone (150 mg kg⁻¹) in the marginal ear vein. Three rabbits were used for this work. All animals have been kept with free access to food and water prior to use.

**Fibre preparation and mechanical apparatus.** Small bundles (50–100 fibres) dissected from soleus muscle were stored in skinning solution containing 50% glycerol at −20 °C for 3–4 weeks and single fibres were prepared just before the experiment as already described[33,63]. The same procedure was followed in the few additional experiments made on the psoas fibres of the rabbit. The increase of interfilamentary distance following cell membrane permeabilization was reversed by the addition of the osmotic agent Dextran T-500 (4% weight/volume)[33,64–68]. A fibre segment 4–6 mm long was clamped at its extremities by T-clips and mounted between the lever arms of a loudspeaker motor, able to impose steps in length complete within 100 μs[69], and a capacitance force transducer, with resonant frequency 40–50 kHz[70]. The extremities of the fibre were fixed first with a rigor solution containing glutaraldehyde (5% v/v) and then glued to the clips with shellac dissolved in ethanol (8.3% w/v[68,71]). This procedure prevents the sliding of the ends of the fibre segment inside the clips and minimizes the shortening of the activated fibre against the damaged sarcomeres at the ends of the segment during force development. Sarcomere length ($sl$), width ($w$) and height ($h$) of the fibre were measured at 0.5 mm intervals in the 3–4 mm central segment of the relaxed fibre with a 40x dry objective (Zeiss, NA 0.60) and a 25x eyepiece. The fibre length ($L_0$) was adjusted to have a $sl$ of 2.3–2.5 μm. The fibre cross-sectional area (CSA) was determined assuming the fibre cross-section as elliptical (CSA = $\pi/4 \cdot w \cdot h$) and its value, in the presence of 4% dextran, ranged between 1900 and 5800 μm² (3500 ± 1000 μm², mean ± SD from 37 fibres). Fibres were activated by temperature

jump using a solution exchange system[33]. A striation follower[72] allowed nanometer-microsecond resolution recording of length changes in a selected population of sarcomeres (range 500–1200 sarcomeres) starting at the time the optic path was permitted through the glass window in the floor of the test temperature drop (see[33] for details). Data in control are from 31 fibres out of 37 fibres, data in the presence of 1 μM OM from 19 out of 37 fibres. The effect of temperature on the relevant mechanical parameters (Fig. 2 and Fig. 3) has been determined at 12, 17, 20, 25, 30 and 35 °C in control (31 fibres) and at 12, 20, 25, 30 and 35 °C in the presence of 1 μM OM (17 fibres). The force-pCa relations have been determined at 12, 17, 25 and 35 °C in control (19 fibres) and at 12, 25 and 35 °C in the presence of 1 μM OM (11 fibres) (Table 1).

### Force-pCa relation.

The dependence of active isometric force on [Ca$^{2+}$] in demembranated fibres from soleus muscle was estimated in the range of pCa values from 8 to 4.2. The force-pCa relation was interpolated by the Hill equation:[73]

$$T_0/T_{0,4.5} = 1/[1 + 10^{n_H \cdot (pCa - pCa_{50})}] \quad (1)$$

where $T_{0,4.5}$ is the isometric force at saturating [Ca$^{2+}$] (pCa ~4.5), $n_H$, the Hill coefficient, indicates the slope of the relation and thus the degree of cooperativity of thin filament activation, pCa$_{50}$, the pCa value for $T_0 = 0.5 \cdot T_{0,4.5}$, is an estimate of the Ca$^{2+}$ sensitivity of the contractile system. The force-pCa relations were determined at different temperatures (range 12–35 °C) in a sequence varied at random in the absence (control) and in the presence of 1 μM OM.

### Stiffness measurements and estimate of the contributions of myofilaments and myosin motors to half-sarcomere compliance.

Step length changes (ranging from −2 to +2 nm per hs, stretch positive), complete in 110 μs, were imposed on the isometrically contracting fibre (Supplementary Fig. 2a–c) to estimate the half-sarcomere stiffness ($k_0$). To enhance the precision of the measurement, a train of different-sized steps at 200 ms intervals was superimposed on $T_0$ and, to maintain the sarcomere length and $T_0$ constant before the test step, each test step was followed, after a 50-ms pause, by a step of the same size but in opposite direction. $k_0$ is measured by the slope of the relation between the force attained at the end of the step ($T_1$) and the change in half-sarcomere (hs) length recorded by the striation follower[72] in a segment of the fibre ($T_1$ relation, Supplementary Fig. 2d) (see also ref. [68]).

The stiffness of the array of motors in the half-sarcomere ($e_0$) was calculated from $k_0$ using a simplified mechanical model of the half-sarcomere (Model 1, Supplementary Fig. 1), in which the hs compliance ($C_{hs} = 1/k_0$) is defined by the sum of the compliance of the array of motors ($1/e_0$, the reciprocal of the stiffness) and the cumulative equivalent compliance of the actin and myosin filaments ($C_f$):

$$C_{hs} = (1/k_0) = C_f + 1/e_0 \quad (2)$$

Under conditions in which $T_0$ changes in proportion to the number of motors, like in isometric contractions at different pCa[33], the strain of each motor, $s_0$, remains constant and the compliance of the motor array changes in inverse proportion to $T_0$: $1/e_0 = s_0/T_0$ and thus $s_0 = T_0/e_0$. The strain of the half-sarcomere ($Y_0 = C_{hs}*T_0$) can be expressed by the sum of filament strain ($C_f T_0$) and the motor strain. Thus, from Eq. 2:

$$Y_0 = (C_{hs} \cdot T_0 = C_f \cdot T_0 + T_0/e_0 =)C_f \cdot T_0 + s_0 \quad (3)$$

In this case (i) the slope and the ordinate intercept of the first order equation fit to the half-sarcomere strain-force ($Y_0 - T_0$) relation (Supplementary Fig. 3b) estimate $C_f$ and $s_0$ respectively and (ii) the stiffness of the array of myosin motors ($e_0$) increases in proportion to the Ca$^{2+}$-dependent increase in $T_0$ (Supplementary Fig. 3c), underpinning a proportional increase in the fraction of attached motors.

For $T_0 < 50$ kPa the $Y_0 - T_0$ data are shifted downward with respect to the linear fit to data ≥50 kPa (continuous line in Supplementary Fig. 3b). This deviation from linearity at very low forces has been explained with the presence of an elastic element in parallel with the myosin motors (Model 2 in Supplementary Fig. 1) with a compliance $C_p$ that is one order of magnitude larger than the compliance of the array of motors at saturating [Ca$^{2+}$]. Consequently, the $Y_0$-$T_0$ relation is affected only in the region where the number of the motors (and thus their cumulative stiffness) is so low to become comparable to the stiffness of the parallel elastic element[35,74]. The contribution of this parallel elasticity emerges also as an upward deviation of the $e_0 - T_0$ relation for forces <50 kPa (Supplementary Fig. 3c). In this respect it has been demonstrated that the first order equation fit to data for forces ≥50 kPa (continuous lines in Supplementary Fig. 3b, c) provides estimates of $s_0$, $C_f$ and $e_0$ that are not significantly affected by the parallel elasticity[35,74]. Notably, the $Y_0 - T_0$ relation in 1 μM OM (Supplementary Fig. 3d) does not show deviation from linearity for the same low forces at which the control relation shows the downward shift, because in this case the number of attached motors and thus their cumulative stiffness are higher than in control[30].

### Data collection and analysis.

Force, motor position and sarcomere length signals were recorded with a multifunction I/O board (PXIe-6358, National Instruments). A program written in LabVIEW (National Instrument) was used for signal generation and data acquisition. All data were analysed using dedicated programs written in LabVIEW (National Instruments) and Microsoft Excel and Origin 2018 (OriginLab Corp., Northampton, MA, USA) software.

### Statistics and reproducibility.

For the slow and fast fibres in the control and for the slow fibres in the presence of 1 μM OM the statistical significance of the temperature effect is tested with one-way ANOVA. Error bars on mean data points are ± SEM.

### Solutions.

The composition of the solutions (Supplementary Table 1) was calculated with a computer program similar to that described in ref. [75] and[63] by taking into account the effect of temperature on the equilibrium between two ions (Ca$^{2+}$ and Mg$^{2+}$) and two chelators (EGTA and ATP) by using the free available Max-chelator software (https://somapp.ucdmc.ucdavis.edu/pharmacology/bers/maxchelator/CaMgATPEGTA-NIST-Plot.htm) developed by Dr. Chris Patton (see also[76]). Due to the temperature dependence of the pK of the buffer used, the total concentration of TES decreases with the increase in temperature from 140 mM (at 5 °C) to 40 mM (at 35 °C)[77]. Free Mg$^{2+}$ and MgATP were in the range 1.7–1.9 mM and 4.9-5.0 mM respectively, with 190 mM ionic strength. Cysteine and cysteine/serine protease inhibitors (trans-epoxysuccinil–L–leucylamido–(4–guanidine) butane, E–64, 10 μM; leupeptin, 20 μg/ml) were also added to all solutions, in order to preserve lattice proteins and thus sarcomere homogeneity. The activating solution at a given pCa (range 8-4.2) was obtained by mixing relaxing and activating solution. OM (CK-1827452, Selleckchem) was dissolved in DMSO (Sigma D-5879) to obtain a 17.5 mM stock solution. OM concentration in the final solutions (1 μM) was obtained by dilution starting from the stock solution[30]. 4% dextran T-500 (Thermo Fisher Scientific) was added to all solutions to restore the lattice spacing before skinning[33,35,64,65].

### Reporting summary.

Further information on research design is available in the Nature Research Reporting Summary linked to this article.

## Data availability

The authors declare that the data supporting the findings of this study are available within the paper and its Supplementary Information files. The source data for Figs. 2, 3, 4, Supplementary Figs. 2, 3, 4, 5 and 6 are provided as a Supplementary Data 1 excel file. All remaining data will be available from the corresponding author upon reasonable request.

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

## Acknowledgements

We thank Corrado Poggesi for critical evaluation of this work. We thank the staff of the mechanical workshop of the Department of Physics and Astronomy (University of Florence) for mechanical engineering support. This work was supported by the University of Florence, competitive project marcocaremani_rictd1819 (Italy), Fondazione Cassa di Risparmio di Firenze (2018.0033 and 2020.1582 (Italy)) and the Italian Ministry of Education, Universities and Research (DR 1647 and DR 1638) under the European Joint Programme on Rare Diseases (IDOLS-G, EJPRD19-126 and PredACTINg, EJPRD19-033).

## Author contributions

M.L., V.L., G.P., M.C., and M.R. designed the research. M.C., M.M., I.M., P.B., I.P. and C.S. performed the experiments and analysed the data. V.L., M.L., G.P. wrote the paper or revised it critically for important intellectual content. All authors participated in discussions on this work and approved the final version of the manuscript.

## Competing interests

The authors declare no competing interests.
