## [Peer Review File · Communications Biology]

Reviewers' comments:

Reviewer #1 (Remarks to the Author):

This manuscript investigates the molecular mechanisms that modulate cooperativity in skeletal muscle. The authors present new data from sarcomere-level mechanical analysis of skeletal fibers measured at different temperatures and in the absence/presence of a myosin-targeting myotrope. The main conclusion is that cooperativity depends on the average force generated by a myosin head.

Cooperativity is an important facet of muscle performance but remains poorly understood. A paper that unveils new molecular mechanisms that contribute to cooperativity could have a very high impact.

The experiments and the analysis of the data are of very high quality. The conclusion that cooperativity is proportional to the force per myosin head is novel and intriguing. These are clear strengths of the work.

The model is less convincing, largely because it is presented only in qualitative detail. The mechanism proposed by the authors (that thin filament spread depends on cross-bridge force) may fit the data but it is not tested in detail and/or with mathematical rigor.

Specific questions

How does cooperativity scale with absolute fiber force? (Only the relationship between cooperativity and force per head is shown)

How does thick filament activation influence cooperativity in the authors' scheme? In prior work, the authors have led the world by showing that stress increases the number of heads in the ON / DRX configuration. If these additional DRX heads attached to the thin filament, they could contribute to thin filament activation and cooperativity. This seems a simpler mechanism through which force could enhance cooperativity. It should be possible to test this concept moderately easily with computational models.

Minor

The text, while always scientifically clear, could benefit from additional editing to improve grammar and style.

Reviewer #2 (Remarks to the Author):

The manuscript COMMSBIO-22-1387-T "The force of the myosin motor sets cooperativity in thin filament activation of skeletal muscle" by Caremani et al describes evidence that the force, not number of myosin heads attached to a muscle myofilament (sarcomeric) thin filament, drives cooperativity. Using permeabilized rat skeletal muscles, the authors also describe that non-force generating myosin heads may inhibit cooperativity. Overall, the authors describe an elegant study, but some concerns exist:.

1. Evidence regarding OM causing drag.

Overall, the authors provide a logical description of why OM-induced drag on the thin filament might alter n_H . Could this be further explained by the state of the myosin ATPase cycle that these heads lie? (I.e. are the drag-inducing heads in pre-power stroke phase only?)

1.a. These findings appear to be based on the assumption that the "strain of each motor, s_0 , remains constant", at least in isometric conditions (Methods, ~Line 390). However, in Ref29, the authors show

that s is temperature dependent. If s_0 is not considered constant, how would this alter the conclusions of the study?

1.b. Discussion, Lines 277-291. Are the authors aware of any evidence that the fast TnC isoform alters the number of attached heads and/or alters myosin binding in a manner that would be consistent with their findings regarding OM and its alteration of n_H ? i.e. does the transition from fast TnC to slow/cardiac TnC also alter the number of attached heads?

2. Statistics

2.a. r - and p -values should be included for regression relationships when multiple temperatures or fibers are used.

2.b.. Please clarify if the regressions were simple linear regressions based on averages per fiber or multiple linear regressions/mixed models, which might better control for variation between preparations.

3. Minor Suggestions

3.a. Please include a description of k_0 , Y_0 , etc in the figure legends. (i.e. " k_0 , half sarcomere stiffness")

3.b. T_1 is not well described until the methods (line 380). Should be described around Line 108 (and the figure legends if appropriate).

3.c. Figures 2,3 D-F, it is difficult to discern what temperatures are used. Please clarify that there are 6 temperatures as described in the Results (~Line 128). Please also indicate which is at the lowest and which is at the highest temperatures, potentially via color as done in Figure 4.

3.d. Line 281. Do the authors mean "complicated" rather than "complicate"?

Reviewer #3 (Remarks to the Author):

This is an interesting paper that attempts to account for the changes in the force - pCa relationships seen at different temperature by a series of very careful mechanical measurements and some explanatory modelling of the data.

The mechanical data collection is detailed and the arguments presented are for the main part plausible (see below) but the work depends upon a body of work going back to the classical work of Huxley Simmons which may not be familiar to everyone not directly involved in mechanical measurements. This makes for a difficult read for anyone not fully familiar with the background work and will limit the potential audience. It would help to have the assumption underlying the model to be clearly stated. The work will therefore be of primary interest to those active in the field. There are some issues that need to be addressed.

Detailed critique.

1. Force dependent regulation of the thick filament is believed to have a role in defining the cooperativity in the force pCa plots but is not discussed here. The Cooperativity in the thin filament is much lower in the absence thick filaments, i.e. n is less than or equal to 2.

2. The value of n is highly variable in measurements from different laboratories. What accounts for this variability and does that affect the work here?

3. The thick filament is more ordered/off at high temperature, does this affect the temperature dependence of n .

4. The equilibrium between blocked and open & closed states is also temperature dependent with fewer blocked states at high temperature. Could this affect the detailed interpretation (Head et al 1999, Eur J Biochem).

5. The work is interpreted through the 3-state/ steric blocking model of thin filament regulation (ref 17) which raises a question. Does force per motor equate to actin binding energy per motor? If so, then this the model is similar in principle to the original 3-state model of actin regulation in which the free energy change of the Tm movement between closed and open positions (equilibrium const. K_T) is

coupled to the free energy change of myosin going from weakly bound to strongly bound (equilibrium constant, K_2).

6. Fig 4C combines two data sets from soleus and psoas fibers to produce the hyperbola n vs F plot. There is no curvature in the soleus data. It is hard to assess if there is any evidence of curvature in the Psoas data. Is the curvature just produced by combining two st line plots? The double reciprocal plot does not resolve the issue.

7. The interpretation of the OM data by Ostap and Goldmann suggests that OM inhibits the working stroke size and at the same time decreases the detachment rate prolonging cross bridge attached lifetime. The longer attachment lifetimes accounting for the activation of the thin filament by OM. Is your data compatible with this model? Either stroke size or attachment lifetime might be expected to alter the average force per myosin in the steady state.

8. Is the effect of OM on soleus exactly the same as on Psoas?

We thank the Editor and the Reviewers for their comments and suggestions (reported point by point in *italic*), which allowed us to improve the quality of the presentation and better clarify some critical issues also with the addition of new data.

Reply to the Reviewer #1

This manuscript investigates the molecular mechanisms that modulate cooperativity in skeletal muscle. The authors present new data from sarcomere-level mechanical analysis of skeletal fibers measured at different temperatures and in the absence/presence of a myosin-targeting myotrope. The main conclusion is that cooperativity depends on the average force generated by a myosin head.

Cooperativity is an important facet of muscle performance but remains poorly understood. A paper that unveils new molecular mechanisms that contribute to cooperativity could have a very high impact.

The experiments and the analysis of the data are of very high quality. The conclusion that cooperativity is proportional to the force per myosin head is novel and intriguing. These are clear strengths of the work.

The model is less convincing, largely because it is presented only in qualitative detail. The mechanism proposed by the authors (that thin filament spread depends on cross-bridge force) may fit the data but it is not tested in detail and/or with mathematical rigor

As it is made explicit in introducing the related section (A mechanistic model that explains the dependence...), the model is considered as “a proof-of-context to qualitatively materialise as the discovery of the role of force per motor in thin filament cooperative activation can be integrated in the current steric-blocking model”. The scope is to prove that the force of the motor can explain both the amplification and the drug dependent inhibition of cooperativity and trigger a quantitative elaboration based on the findings in the present experimental work, which, as acknowledged by the reviewer, finds its strength in unveiling a new molecular mechanism for cooperativity.

How does cooperativity scale with absolute fiber force? (Only the relationship between cooperativity and force per head is shown)

The rationale for the experiment is to test the effect of the motor force on the cooperativity of thin filament activation. For this, we selected protocols that modulate the fibre force by “only” changing the force per motor F_0 at constant number of attached motors. As demonstrated in published work on fast skinned muscle fibres (Linari et al. Biophys J. 2007, 92:2476) and reported here for slow skinned fibres, this condition can be pursued under sarcomere level mechanical control by changing temperature of the Ca-activated fibre. The evidence is given by the direct proportionality between the strain of the hs and the fibre force modulated by temperature at pCa 4.5 ($T_{0,4.5}$) (Fig. 5 filled circles in Linari et al. 2007; Fig. 2D here). Once the compliance of the filaments is taken into account, the results hold the conclusion that the average strain of the motor (s_0) and thus F_0 are directly proportional to the fibre force ($T_{0,4.5}$) and that the number of attached motors remains constant (Fig. 2F this work). In other words, the increase in $T_{0,4.5}$ with temperature is totally explained by increase in F_0 . Consequently, it looked redundant to us to use a second abscissa ($T_{0,4.5}$, calculated from the fit of the $F_0 - T_{0,4.5}$ relation) in the panel showing the relation of n_H versus F_0 (Fig. 4A). Anyway, we are ready to add the second abscissa (see Fig R1, attached) if the Reviewer considers it necessary. Obviously, the same argument cannot be applied when we extend the relation to n_H data from fast skinned fibres (Fig. 4C), because in this case T_0 depends also on the number of motors that is different for the different isoforms.

How does thick filament activation influence cooperativity in the authors' scheme? In prior work, the authors have led the world by showing that stress increases the number of heads in the ON / DRX configuration. If these additional DRX heads attached to the thin filament, they could contribute to thin filament activation and cooperativity. This seems a simpler mechanism through which force could enhance cooperativity. It should be possible to test this concept moderately easily with computational models.

The question about the contribution of mechanosensing-based myosin filament activation to the present results and their interpretation was not discussed in the paper as we relied on the evidence that thick filament activation does not give a significant contribution under the conditions of these experiments. However, following the question of both Reviewers 1 and 3, this evidence, reported below, is discussed in the revised version. An indispensable procedure to maintain sarcomere length homogeneity during force development of skinned fibres is the temperature jump technique that requires that the activating solution is first equilibrated into the fibre at 1°C (Linari et al. Biophys J. 2007, 92:2476). As demonstrated in Caremani et al. J. Gen Physiol. 2021, 153 e202012713, the thick filament structure of skinned fibres has already undergone a significant transition to the ON/disordered state at 10 °C and presumably this transition is even more effective at 1 °C. In addition, the absence of a significant effect of myosin filament activation in the present experiments is suggested by the lack of asymmetry in the force pCa relation across the midpoint. In fact, while thin filament activation is complete only with Ca²⁺ saturation at the maximum force ($T_{0,4.5}$), mechanosensing-based myosin filament activation is complete when the force is nearly half-maximum (Piazzesi et al. Front Physiol. 2018, 9:736). Thus, myosin filament should contribute to the shape of the first half of the force-pCa relation, making the whole relation asymmetric, which is excluded by the present data. The results of this paper are due to a refined experimental approach that allows to reveal the n_H - F_0 relation and we appreciate the expectation of the Reviewer that they foster future modelling work.

Minor

The text, while always scientifically clear, could benefit from additional editing to improve grammar and style.

Text improved especially concerning the reduction of long sentences.

Reply to the Reviewer #2

The manuscript COMMSBIO-22-1387-T "The force of the myosin motor sets cooperativity in thin filament activation of skeletal muscle" by Caremani et al describes evidence that the force, not number of myosin heads attached to a muscle myofibril (sarcomeric) thin filament, drives cooperativity. Using permeabilized rat skeletal muscles, the authors also describe that non-force generating myosin heads may inhibit cooperativity. Overall, the authors describe an elegant study, but some concerns exist:.

1. Evidence regarding OM causing drag.

Overall, the authors provide a logical description of why OM-induced drag on the thin filament might alter n_H . Could this be further explained by the state of the myosin ATPase cycle that these heads lie? (I.e. are the drag-inducing heads in pre-power stroke phase only?)

The question by the reviewer has been directly answered in previous work, as referred in the paper. The direct evidence that attached OM-heads are arrested at the beginning of the working-stroke comes from the single molecule mechanical measurements of Woody et al., Nature Communications, 2018, 9:3838. In a previous paper we determined in situ stoichiometry of the OM-head interaction in slow muscle fibres and found that with 1 μM OM the force of the Ca^{2+} activated fibre is halved while the number of attached motors is the same as in control (Governali et al. Nature Communications. 2020, 11:3405), concluding that one half of the attached motors had OM bound and, as demonstrated by Woody et al. 2018, did not go through the working stroke.

1.a. These findings appear to be based on the assumption that the “strain of each motor, s_0 , remains constant”, at least in isometric conditions (Methods, ~Line 390). However, in Ref29, the authors show that s is temperature dependent. If s_0 is not considered constant, how would this alter the conclusions of the study?

Whether s_0 and thus F_0 are constant or not is not the result of assumptions but directly depends on the protocol used to modulate T_0 , either changing activating $[\text{Ca}^{2+}]$ or temperature.

1. Changing $[\text{Ca}^{2+}]$ at a given temperature to generate the force-pCa relation: as demonstrated in previous work for fast skeletal muscle fibres (Linari et al. 2007) and in this work for slow fibres (Supplementary Information and specifically Supplementary Fig. 3), the isometric force developed under these conditions is related to the number of attached motors that depends on $[\text{Ca}^{2+}]$, while the strain s_0 (and the force F_0) of each attached motor is constant independent of $[\text{Ca}^{2+}]$.
2. Changing temperature at constant $[\text{Ca}^{2+}]$: in this way the isometric fibre force is changed by changing s_0 (and thus F_0) without changing the number of motors (Suppl Fig. 4 and Fig. 2). This is the protocol of election in the present work for studying the effect of changing the force per motor on the steepness of the force-pCa relation.

1.b. Discussion, Lines 277-291. Are the authors aware of any evidence that the fast TnC isoform alters the number of attached heads and/or alters myosin binding in a manner that would be consistent with their findings regarding OM and its alteration of n_H ? i.e. does the transition from fast TnC to slow/cardiac TnC also alter the number of attached heads?

The question appears to us not to be consistent with the relevant issue in this paper that is the original demonstration that the cooperativity in thin filament activation can be accounted for by the force per motor, independent of the TnC-related Ca^{2+} sensitivity and thus on the $p\text{Ca}_{50}$. A change in $p\text{Ca}_{50}$ as that attributable to the different (slow-fast) TnC isoform per se does not affect the cooperativity, as shown by the unique n_H - F_0 relation for the two different isoforms in Fig. 4C (circles slow, triangles fast). This conclusion is further solidified in the revised version by new data extending the study in fast muscle fibres to a lower temperature (6 °C) and showing that n_H is the same for the same F_0 value, attained in slow fibre at 35 °C (see revised Table 1). In this view, as concluded in the paper, the questions raised by the complex and often contradictory results found in the literature in relation to TnC substitution could be faced by new experiments that exploit the half-sarcomere compliance analysis shown in this paper to test whether and how the replacement of the regulatory proteins affects the force of the myosin motor.

OM, in contrast to TnC isoform, specifically inhibits the cooperativity and we provide a key for explaining the phenomenon in the example described in the mechanistic model (Fig. 5C-D).

2. Statistics

2.a. r- and p-values should be included for regression relationships when multiple temperatures or fibers are used.

Thanks to the reviewer's suggestion we improved the quantitative aspects of the analysis throughout the paper. We added the parameters of the regression equations and their errors in legends of Fig. 2, 3, 4 and Suppl. Fig. 3 and 4

2.b.. Please clarify if the regressions were simple linear regressions based on averages per fiber or multiple linear regressions/mixed models, which might better control for variation between preparations.

We added the required statistical information to regression lines in Fig. 2D-E and 3D-E

3. Minor Suggestions

3.a. Please include a description of k_0 , Y_0 , etc in the figure legends. (i.e. " k_0 , half sarcomere stiffness")
Done

3.b. T_1 is not well described until the methods (line 380). Should be described around Line 108 (and the figure legends if appropriate).

T_1 is defined now when it appears for the first time in Fig. 2C and at higher resolution in Suppl. Fig. 2C.

3.c. Figures 2,3 D-F, it is difficult to discern what temperatures are used. Please clarify that there are 6 temperatures as described in the Results (~Line 128). Please also indicate which is at the lowest and which is at the highest temperatures, potentially via color as done in Figure 4.

The direction of the temperature change is now indicated by arrows in both Fig. 2D-F (six temperatures) and 3D-F (five temperatures). The temperatures used in each experiment are specified in Methods.

3.d. Line 281. Do the authors mean "complicated" rather than "complicate"?

Amended

Reply to the Reviewer #3

This is an interesting paper that attempts to account for the changes in the force - pCa relationships seen at different temperature by a series of very careful mechanical measurements and some explanatory modelling of the data.

The mechanical data collection is detailed and the arguments presented are for the main part plausible (see below) but the work depends upon a body of work going back to the classical work of Huxley Simmons which may not be familiar to everyone not directly involved in mechanical measurements. This make for a difficult read for anyone not fully familiar with the background work and will limit the potential audience. It would help to have the assumption underlying the model to be clearly stated.

Please note that the relevant parameters of a simplified half-sarcomere mechanical model and the experimental protocols necessary to define them through the half-sarcomere compliance analysis are described in detail in a dedicated section of Methods and in four supplementary Figures.

The work will therefore be of primary interest to those active in the field

There are some issues that need to be addressed.

Detailed critique.

1. Force dependent regulation of the thick filament is believed to have a role in defining the cooperativity in the force pCa plots but is not discussed here.

The question about the contribution of mechanosensing-based thick filament activation to the present results and their interpretation was not discussed in detail in the previous version of the paper, as we relied on the evidence that the thick filament activation did not give a significant contribution in control under the conditions of these experiments. Following the question of both Reviewers 1 and 3, this evidence, reported below, is discussed in detail in the specific section of the revised version and the different motor states are better clarified also in relation to the simplified graphical representation in Fig. 5. An indispensable procedure to maintain sarcomere length homogeneity during Ca^{2+} activated force development at a given temperature is the temperature jump technique that requires that the activating solution is first equilibrated into the fibre at 1 °C (L nari et al. Biophys J. 2007, 92:2476). As demonstrated in Caremani et al. J. Gen Physiol. 2021, 153 e202012713, the thick filament structure of skinned fibres has already undergone a significant transition to the ON/disordered state at 10 °C and presumably this transition will be even more effective at 1 °C, preventing the mechanosensing mechanism to take place. On the other end evidence for the absence of a significant effect of thick filament activation in the present experiments is the lack of any asymmetry in the force pCa relation across the midpoint. In fact, while the thin filament activation is complete only with Ca^{2+} saturation at the maximum force ($T_{0,4.5}$), mechanosensing-based thick filament activation is complete when the force is nearly half-maximum (Piazzesi et al. Front Physiol. 2018, 9:736). If thick filament mechanosensing was present would have contribute to shape the first half of the force-pCa relation, making it asymmetric, which is excluded by the present data. In the graphical simplified representation in control (Fig. 5A-B) it is shown that $\frac{1}{4}$ of motors can attach once the actin monomer is activated. This fraction is close to the duty ratio classically found in steady isometric contraction, in agreement with the assumption that, also at the low Ca^{2+} levels of the simulation, the thick filament activation is nearly maximum.

The Cooperativity in the thin filament is much lower in the absence thick filaments, i.e. n is less than or equal to 2.

The reviewer is right, in the absence of the myosin filament there cannot be any myosin contribution to cooperativity in thin filament activation, whatever the motor force.

2. The value of n is highly variable in measurements from different laboratories. What accounts for this variability and does that affect the work here?

A prerequisite for the study carried here (which requires the precise control of F_0 as an independent variable) is the preservation of sarcomere homogeneity during skinned fibre activation to control mechanics at the level of a selected population of sarcomeres. The required protocols are demanding and not very popular and this can explain the contribution to the variation of the results in the literature. A second level of variation is likely related to the steepness of the $[\text{Ca}^{2+}]$ dependence of the mechanical responses, which requires that data to be compared in the most reliable way for the effect of a given intervention must originate from the same batch of solutions to minimise the effect of even very small variation in $[\text{Ca}^{2+}]$.

We have mentioned in Discussion the complexity of interpreting data in which native proteins are substituted with other isoforms or with mutated proteins in the absence of a refined method for half-sarcomere mechanics.

The thick filament is more ordered/off at high temperature, does this affect the temperature dependence of n.

A generic effect of temperature on n_H is contradicted in Fig. 4 C by the evidence that n_H follows a unique relation with F_0 , independently of temperature. This is further solidified in the revised version of the paper by extending the n_H - F_0 relation for fast muscle fibres to 6 °C, at which the n_H - F_0 point coincides with that for slow fibres at 35 °C

3. *The equilibrium between blocked and open & closed states is also temperature dependent with fewer blocked states at high temperature. Could this affect the detailed interpretation (Head et al 1999, Eur J Biochem).*

In this work (and in previous works of others, see ref. 20) the threshold for the force-pCa relation is not affected by temperature. This indicates that the temperature dependence of the equilibrium between the states hypothesised in the steric-blocking model defined with solution kinetic experiments does not affect per se the position of the force-pCa relation and thus the Ca^{2+} sensitivity in fibre. It seems difficult to see how the claimed effect in solution reflects on mechanical data in fibre demonstrating that the force per motor accounts for the modulation of n_H . That temperature per se is not a determinant of cooperativity, or at least it is not able to influence the n_H - F_0 relation, is made even more evident in the revised version of the manuscript by the coincidence of the n_H - F_0 point for soleus at 35 °C and psoas at 6 °C.

The work is interpreted through the 3-state/ steric blocking model of thin filament regulation (ref 17) which raises a question. Does force per motor equate to actin binding energy per motor? If so, then this the model is similar in principle to the original 3-state model of actin regulation in which the free energy change of the Tm movement between closed and open positions (equilibrium const. KT) is coupled to the free energy change of myosin going from weakly bound to strongly bound (equilibrium constant, K2).

We found sensible to fit the interpretation of our finding into the scheme of the steric-blocking model, but we find it difficult to equate the weak-strong transition defined in solution kinetics or in ionic strength-dependent experiments to the different force generating states defined in our mechanical experiments in physiological conditions, in which the stiffness of the actin-myosin bond is the same at different forces. On the other end, it is clear that (even if a detailed molecular explanation is missed) the dynamic element, identified in the force, required to integrate the original model, must certainly satisfy the energetic requirements for inducing progressively larger Tm movements.

6. *Fig 4C combines two data sets from soleus and psoas fibers to produce the hyperbola n vs F plot. There is no curvature in the soleus data. It is hard to assess if there is any evidence of curvature in the Psoas data. Is the curvature just produced by combining two st line plots? The double reciprocal plot does not resolve the issue.*

The interpretation of the soleus data as the lower branch of a hyperbola is a plausible hypothesis that cannot be discarded just because the analysis cannot be extended to temperatures above 35 °C. The objection of the reviewer has represented a challenge that we found possible to face because the n_H - F_0 relation for fast fibres could be extended to lower temperatures (up to 5 °C), preserving F_0 as the only factor modulated by temperature without changes in the number of motors (Linari et al. 2007). Extending the fast fibre analysis to this low temperature we obtained full overlap of n_H points from the two fibre types for the same F_0 , which further solidifies the role of F_0 and excludes other generic effects of temperature.

7. *The interpretation of the OM data by Ostap and Goldmann suggests that OM inhibits the working stroke size and at the same time decreases the detachment rate prolonging cross bridge attached lifetime. The longer attachment lifetimes accounting for the activation of the thin filament by OM. Is your data compatible with this model? Either stroke size of attachment lifetime might be expected to alter the average force per myosin in the steady state.*

As reported by Woody et al. with single molecule mechanics (Woody et al. Nature Communications. 2018, 9:3838) and by Governali et al. with sarcomere level mechanics in skinned soleus fibres (Governali et al. Nature Communications. 2020, 11:3405), the ATPase rate (and the detachment rate) at high load (nearly isometric conditions) is not affected by OM. Thus, in isometric conditions OM does not induce any increase in the fraction of attachment lifetimes or in the fraction of attached heads. The findings that OM-bound motors do not go through the force generating step and that in fibres in 1 μM OM half of the motors is OM-

bound (Governali et al. 2020) make OM a candidate for testing the effect of average force per motor on cooperativity, as alternative to the temperature. In fact, in the presence of OM the average force is halved by partition between 50% OM-motors with zero force and 50% motors with full force, in contrast to the effect of temperature that changes the force of the motor by changing the equilibrium between force generating states. The dual effect of OM on the force-pCa relation is accounted for in detail in the paper.

8. Is the effect of OM on soleus exactly the same as on Psoas?

The OM protocol was specifically selected for its effects on the force and on n_H of the muscle fibres with the slow myosin isoform. As reported in the literature (Nagy et al. Br J Pharmacology. 2015, 172:4506) its effects on fibres with the fast isoform are much smaller and a specific study of this phenomenon was not in the scope of this work.

Reviewers' comments:

Reviewer #1 (Remarks to the Author):

In their revision, the authors added a little information about the statistical results and added a brief discussion about the potential contribution of transitions from the OFF to ON state of myosin. They did not, as requested, show new data demonstrating how cooperativity scales with absolute fiber force.

The authors state that the myosin heads should already be in the ON state under their experimental conditions (before the T-jump) and thus that OFF/ON transitions will not contribute to the data. However, they do not show this. They also do not show that all heads remain in the ON state after the T-jump.

Additionally, the authors argue that contributions from OFF/ON transitions would alter the symmetry of the force-pCa relationships. They state that their own curves are symmetric, but they don't test this. This is a problem because other groups have shown asymmetrical force-pCa relationships, albeit perhaps under different experimental conditions.

In short, the authors' claims may be correct but their hypothesis is not tested in rigorous detail. Potential alternative interpretations are not given sufficient consideration.

Reviewer #2 (Remarks to the Author):

The revised manuscript COMMSBIO-22-1387-A "The force of the myosin motor sets cooperativity in thin filament activation of skeletal muscle" by Caremani et al is improved in its content to better describe how the force of an attached myosin head drives cooperative activation of the sarcomeric thin filament.

The authors have addressed the major concerns. Minor concerns/suggestions are listed here, which may continue to improve the manuscript.

Figure 5. Could the On/Off state of myosin be better represented here? Presumably, there should be more myosin heads in A that are in the disordered state than B, due to the temperature dependence discussed in the manuscript.

The authors may want to also revise their text to be more consistent with the terminology of on/off versus disordered/helical.

Top of page 8 (tracked version). The word "anticipates" which is the first word on the page appears to be misused. I.e. there is no a priori reason for the OM-bound myosin head to prepare for myosin binding. Perhaps it prepares or primes the filament?

Bottom of page 8/top of page 9 (tracked version). What do the authors mean by threshold? Do the authors mean minimum calcium activation? (i.e. n_H could maintain pCa_50, but cannot if the calcium level at which the force deviates from passive is maintained.)

Table 1. Was statistical testing performed on these data? Statistical results could be of value to many muscle researchers evaluating the data.

Reviewer #3 (Remarks to the Author):

This revised manuscript has addressed all of the substantive issues raised at review. The work presents a self consistent interpretation of the experimental data from the mechanical assays and as such is ready to be published. The changes made to the ms, including the new data strengthen the arguments presented and clarify the presentation.

I remain of the view that there may be more to this story, specifically how Pi release (control or inhibited by OM) is coupled to the cooperativity and the long running question about when Pi is released - before or after force generation.

Reply to the Reviewers

We thank the Editor and the Reviewers for remarking their critical issues, which promoted further revision of the paper with the addition of new analysis (Supplementary Figs 5 and 6 and supplementary Table 2), that better demonstrates the unique power of our approach. Here following we reply point by point to Reviewers' comments (in italic).

Reviewer #1

In their revision, the authors added a little information about the statistical results and added a brief discussion about the potential contribution of transitions from the OFF to ON state of myosin. They did not, as requested, show new data demonstrating how cooperativity scales with absolute fiber force.

The question “*how cooperativity scales with absolute fiber force*” is now explicitly discussed based on the peculiarity of our methodological approach able to define the force per motor and the consequent constraints that this parameter poses on the interpretation of the results. Changes in isometric fibre force (T_0) of a given fibre type due to changes in temperature can be totally accounted for by changes in force per motor and consequently the $n_H - T_0$ relation holds just like the $n_H - F_0$ relation (see for instance soleus fibre relations in supplementary fig. 6). Changes in T_0 due to interventions like increase in Pi or Vi that change the number of attached motors without significant effects on F_0 (Caremani et al., 2008; 2011), are not expected to change n_H . This conclusion is essentially confirmed by data in literature for either Pi (Debold et al., 2006) or Vi (Martyn et al., 2007). This point is now explicitly discussed in the present revised version of the paper.

The authors state that the myosin heads should already be in the ON state under their experimental conditions (before the T-jump) and thus that OFF/ON transitions will not contribute to the data. However, they do not show this. They also do not show that all heads remain in the ON state after the T-jump.

The request “*to show myosin heads in the ON state*” looks not justified to us on the basis of previous specific work and the aims of this work. The structural evidence that myosin head activation does not influence the force-pCa relation because thick filament is mostly already activated is given in previous X-ray work showing that at the low temperature preceding the temperature jump most of the heads are already disordered (Caremani et al., 2021). The finding that the force-pCa relation is symmetric across the force midpoint (see following point for its demonstration) is strictly consequent to this condition. According to Campbell analysis and modelling (1997) a symmetric relation is explained by the presence of only one cooperativity mechanism in thin filament activation and the absence of a significant role for other cooperativity mechanisms operating on part of the relation as it would occur in the presence of myosin filament activation. Low temperature X-ray data in relaxed fibres in Caremani et al. (2021) give full explanation of why the myosin filament activation mechanism is blunted.

Additionally, the authors argue that contributions from OFF/ON transitions would alter the symmetry of the force-pCa relationships. They state that their own curves are symmetric, but they don't test this. This is a problem because other groups have shown asymmetrical force-pCa relationships, albeit perhaps under different experimental conditions.

The argument against our statement that in our experiments the force-pCa relation is symmetric is now specifically addressed according to the analysis of Hill plot with the linearization suggested in previous work

(Moss et al., 1983). Independently of temperature, the linearized plot shows a unique slope across the force midpoint (Supplementary Fig 5). Thus, at difference with conclusions in previous work (Moss et al., 1983, 1985; Debold et al., 2006), our data exclude any supplementary element for cooperativity beyond that responsible for thin filament activation. In this respect we introduced a discussion on the possible methodological reasons that cause asymmetry in the force-pCa relations, specifically reducing the slope for the forces above the midpoint.

In short, the authors' claims may be correct but their hypothesis is not tested in rigorous detail. Potential alternative interpretations are not given sufficient consideration.

Reviewer's critical issues promoted the punctual integrations reported above that we believe allow a better appreciation of the unicity of our results and raise the impact of the paper in the context of the existing literature. For this we thank the Reviewer #1.

Reviewer #2

The revised manuscript COMMSBIO-22-1387-A "The force of the myosin motor sets cooperativity in thin filament activation of skeletal muscle" by Caremani et al is improved in its content to better describe how the force of an attached myosin head drives cooperative activation of the sarcomeric thin filament. The authors have addressed the major concerns. Minor concerns/suggestions are listed here, which may continue to improve the manuscript.

Figure 5. Could the On/Off state of myosin be better represented here?

The reviewer is right, there is a given degree of simplification in the representation of the state of myosin motors that at first glance may be confusing. As described in the text, for the sake of simplicity, the conventional representation of the ON state (disordered, away from the surface of the filament) is used here also for indicating the motors that are able to attach on the nearby actin monomer once it is activated. A more strictly complete representation should discriminate within motors ON (disordered) those able and those unable to attach according the duty ratio. This further distinction would make the interpretation of the draft too complicate.

Presumably, there should be more myosin heads in A that are in the disordered state than B, due to the temperature dependence discussed in the manuscript.

According to our experimental protocol, in either A and B the test temperature is attained by temperature jump from 1 °C. As discussed in relation to reviewer # 1 question, this is why the degree of thick filament activation appears not to have a role in force - pCa relation. Please note that the temperature dependence of the contracting fibre shown in the paper concerns the force of the motor.

The authors may want to also revise their text to be more consistent with the terminology of on/off versus disordered/helical.

We think that to a first approximation, from structural point of view, OFF is equivalent to ordered and ON is equivalent to disordered (away from the surface of the thick filament). This is clarified in the Introduction.

Top of page 8 (tracked version). The word "anticipates" which is the first word on the page appears to be misused. I.e. there is no a priori reason for the OM-bound myosin head to prepare for myosin binding. Perhaps it prepares or primes the filament?

Please consider that at page 6 we state: "The much larger Ca^{2+} sensitivity characterising the force-pCa relation in OM is in agreement with structural evidence from both fluorescent probes³⁶ and X-ray diffraction³⁷ that OM binding to myosin promotes the release of a fraction of motors from the resting OFF

state even at low $[Ca^{2+}]^i$.

Bottom of page 8/top of page 9 (tracked version). What do the authors mean by threshold? Do the authors mean minimum calcium activation? (i.e. n_H could maintain pCa_{50} , but cannot if the calcium level at which the force deviates from passive is maintained.)

Threshold meaning is now specified the first time it is mentioned (page 7).

Table 1. Was statistical testing performed on these data? Statistical results could be of value to many muscle researchers evaluating the data.

Statistical testing of data in Table 1, done with one-way ANOVA (temperature as independent variable), is now reported in the new Supplementary Table 2.

Reviewer #3

This revised manuscript has addressed all of the substantive issues raised at review. The work presents a self consistent interpretation of the experimental data from the mechanical assays and as such is ready to be published. The changes made to the ms, including the new data strengthen the arguments presented and clarify the presentation.

I remain of the view that there may be more to this story, specifically how Pi release (control or inhibited by OM) is coupled to the cooperativity and the long running question about when Pi is released - before or after force generation.

Stimulated by this observation in the new revision we have added the information from literature about the absence of effect of Pi on n_H and its interpretation.

Reviewers' comments:

Reviewer #1 (Remarks to the Author):

The use of temperature jumps to activate the permeabilized fibers complicates the interpretation of the experimental data and might minimize the effect of a potential recruitment of myosin heads from the OFF or super-relaxed state on cooperativity.

Specifically, the authors use prior publications to justify their claim that all of the myosin heads are in a disordered state, and thus available to attach to actin, at the low temperature prior to the temperature jump. They then step the experimental temperature to a new value and measure the isometric force. These forces are used to construct force-pCa curves from which the Hill coefficients are derived.

Are the force-pCa curves measured with a temperature jump the same as the force-pCa curves measured without a temperature jump? If they are not, the authors may be missing a potential additional contribution to cooperativity. One possibility is that force-pCa curves measured at a fixed temperature have a higher Hill coefficient than curves measured using the temperature jump. The "additional" slope would reflect recruitment from the OFF state.

In summary, while Hill coefficients scale with myosin force in the current experiments, it is possible that this is an unintended artefact associated with the technique used to measure the force-Ca relationship.

Reviewer #2 (Remarks to the Author):

The further revised manuscript COMMSBIO-22-1387-B "The force of the myosin motor sets cooperativity in thin filament activation of skeletal muscle" by Caremani et al provides a model of how the force of an attached myosin head drives cooperative activation of the sarcomeric thin filament. The authors rightfully acknowledge that this is a complicated condition and have addressed all of this reviewer's concerns.

manuscript COMMSBIO-22-1387 B
3rd revision

Reply to Reviewers

Following the last recommendations of the Editor and Reviewer #1 we have integrated and reorganized the Discussion to let the considerations on possible alternative explanations of data suggested by the literature to emerge and be clarified. The modifications can be identified in the yellow highlighted text.